# Supramolecular assembly activated single-molecule phosphorescence resonance energy transfer for near-infrared targeted cell imaging

Xiaolu Zhou[1,3], Xue Bai[1,3], Fangjian Shang[1], Heng-Yi Zhang[1], Li-Hua Wang[1], Xiufang Xu[1] & Yu Liu ®[1,2] ✉

Pure organic phosphorescence resonance energy transfer is a research hotspot. Herein, a single-molecule phosphorescence resonance energy transfer system with a large Stokes shift of 367 nm and near-infrared emission is constructed by guest molecule alkyl-bridged methoxy-tetraphenylethylene-phenylpyridines derivative, cucurbit[n]uril (n = 7, 8) and β-cyclodextrin modified hyaluronic acid. The high binding affinity of cucurbituril to guest molecules in various stoichiometric ratios not only regulates the topological morphology of supramolecular assembly but also induces different phosphorescence emissions. Varying from the spherical nanoparticles and nanorods for binary assemblies, three-dimensional nanoplate is obtained by the ternary co-assembly of guest with cucurbit[7]uril/cucurbit[8]uril, accompanying enhanced phosphorescence at 540 nm. Uncommonly, the secondary assembly of β-cyclodextrin modified hyaluronic acid and ternary assembly activates a single intramolecular phosphorescence resonance energy transfer process derived from phenyl pyridines unit to methoxy-tetraphenylethylene function group, enabling a near-infrared delayed fluorescence at 700 nm, which ultimately applied to mitochondrial targeted imaging for cancer cells.

Supramolecular assembly based on multiple hydrogen bonds[1], halogen bonds[2], metal coordination interactions[3], and macrocycle encapsulation interactions[4,5] have long been a hot topic in molecular recognition[6], catalysis[7–9], luminous materials[10–13], medicine[14–17], and sensing[18,19]. Among them, the macrocyclic supramolecular assembly has garnered considerable attention due to its potent ability to suppress singlet or triplet exciton vibration, which induces and improves the guest photophysical characteristics[20–22], especially the room-temperature phosphorescence (RTP) behavior[23–25]. For example, Tian and coworkers reported a series of assembly-induced efficient amorphous RTP materials via modifying phosphor moieties onto β-cyclodextrin (β-CD)[26]. Our group constructed a cascade-assembly-enhanced phosphorescence system based on cucurbit[8]uril and amphiphilic calixarene for cell imaging[27]. Despite the rapid development of RTP systems in recent years[28–30], achieving tunable phosphorescence emission, especially in the near-infrared (NIR) region still faces great challenges owing to the limitation of the energy-gap law[31]. Notably, phosphorescence resonance energy transfer (PRET) that transfers energy from excited triplet state of ultralong organic RTP emitters donor to singlet state of fluorescent chromophores has been proved to be an efficient path for constructing a long-wavelength and long-lifetime delayed fluorescence to achieve tunable afterglow

[1]College of Chemistry, State Key Laboratory of Elemento-Organic Chemistry, Nankai University, Tianjin, P. R. China. [2]Collaborative Innovation Center of Chemical Science and Engineering (Tianjin), Nankai University, Tianjin, P. R. China. [3]These authors contributed equally: Xiaolu Zhou, Xue Bai.
✉e-mail: yuliu@nankai.edu.cn

emission, which expands the wide application of RTP materials in bioimaging[32], sensing[33,34], and information anti-counterfeiting[35–37]. George et al. proposed delayed sensitization of dye singlet states by the phosphorescence resonance energy transfer of organic phosphor donors to achieve red afterglow fluorescence[38]. Chi and coworkers described a stepwise PRET system utilizing triphenylene-dyes (Nile red, Cyanine 7)-doped polymers, which earned multicolor afterglow anti-counterfeiting[39]. Li et al. reported an intraparticle-PRET-based NIR nanoprobe with the aid of amphiphilic triblock copolymers intended for in vivo afterglow imaging[40]. Besides, supramolecular cascade assembly based on macrocyclic confinement-induced phosphorescence by virtue of noncovalent interaction has been proven to be a potential and convenient strategy for constructing PRET in the aqueous phase, enabling not only efficient light-harvesting systems but also NIR-delayed fluorescence for biosensing[41,42]. However, the most reported PRET systems at present are achieved through doping commercial fluorescent dyes or assembly components as acceptors[43], single intramolecular PRET based on macrocyclic confined guest molecule has been rarely reported to the best of our knowledge.

In this work, an efficient single-molecule PRET system based on macrocyclic confinement and polysaccharide mediation was constructed by alkyl-bridged methoxy-tetraphenylethylene-bromophenylpyridines derivative (TPE-DPY), cucurbit[n]uril(n = 7, 8), and β-cyclodextrin modified hyaluronic acid (HACD), contributing to the targeted cancer cell imaging with a large Stokes shift of 367 nm and NIR emission (Fig. 1). Benefiting from the encapsulation of cucurbituril hydrophobic cavities which effectively promoted the intersystem crossing process and inhibited the non-radiative transition caused by the disorder molecular motion and quenchers, the binary assembly of cucurbit[7]uril (CB[7]) or cucurbit[8]uril (CB[8]) to TPE-DPY all induced a distinct intense phosphorescent emission around 530 nm. By adjusting the ratios of CB[7] and CB[8], the supramolecular co-assembly TPE-DPY/CB[7]/CB[8] exhibited a stepwise enhanced phosphorescence with phosphorescent lifetime extended from 29.09 μs up to 80.64 μs and presented hierarchical self-assembled three-dimensional nanoplates differing from spherical nanoparticles of TPE-DPY/CB[7] and pseudorotaxane nanorods of TPE-DPY/CB[8]. After the further assembly with negatively charged HACD, a single-molecule PRET process derived from phenyl pyridines unit to methoxy-tetraphenylethylene portion was achieved accompanied by the change in topological morphology caused by the confinement effect of β-CD to methoxy-tetraphenylethylene group and electrostatic interaction with HA, ultimately giving an NIR delayed fluorescence at 700 nm with a lifetime recorded as 21.60 μs. Taking advantage of the targeting properties of HACD, the aggregate with a large Stokes shift and long-lived NIR photoluminescence was successfully employed for targeted imaging of cancer cells.

## Results
### Topological morphology and binding behavior of supramolecular assemblies
Two kinds of guest molecules, methoxy-tetraphenylethylene derivatives with one (TPE-PY) or two (TPE-DPY) flexible alkyl-bridged phenyl pyridines groups were synthesized by Mizoroki-Heck reaction and alkyl substitution reaction, which were characterized through nuclear magnetic resonance ($^1$H NMR, $^{13}$C NMR, 2D COSY) and high-resolution mass spectrometry (HRMS) (Supplementary Figs. 1-12). A series of reference molecules, including alkyl-chain-modified bromophenylpyridinium salts (PY-1), tetraphenylethene derivatives possessing vinyl pyridine salts (TPE-1, TPE-2), and alkyl-bridged styrylpyridine-phenylpyridinium derivatives (SP-PY) (Supplementary Figs. 1 and 13–17) were synthesized to process the relevant control experiments for exploring the binding mode of guest molecules with CB[n] (n = 7/8) and the single-molecule PRET

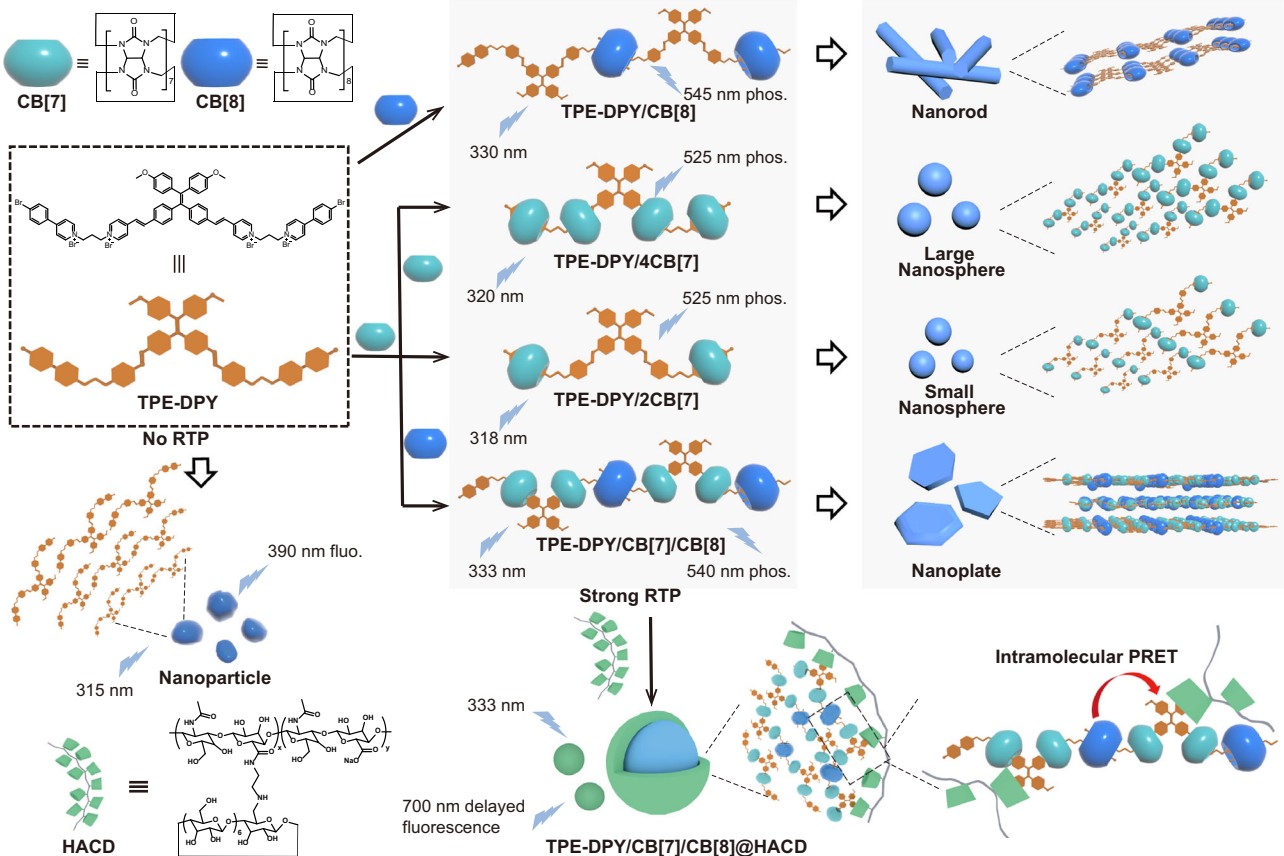

**Fig. 1** | Schematic illustration of the tunable self-assembly mechanism between TPE-DPY, CB[7], and CB[8], as well as the single-molecule PRET process in assembly.

luminescence behavior. The substitution degree of the $\beta$-CD on HA was determined to be 35% (Supplementary Fig. 18). In contrast to the previously reported mono-bromophenylpyridine derivatives[44,45], the guest molecule TPE-DPY not only has two alkyl-bridged bipyridine salt units, which provide more binding sites to assemble with CB[n] ($n$ = 7/8) through ionic dipole interaction and hydrophobic interaction but also has one tetraphenylethene unit a typical AIE molecule with a rigid backbone and hydrophobicity, making it easy for directed assembly in an aqueous solution. First, $^1$H NMR experiments (Supplementary Fig. 19) were conducted to investigate the binding behavior between TPE-DPY and CB[8]. In Supplementary Fig. 19, upon the addition of CB[8] into the guest solution, the proton signal of TPE-DPY gradually passivated and no longer changed when the CB[8] concentration exceeded 1 equivalent, indicating the complexation of TPE-DPY and CB[8] reached an equilibrium stage. Similarly, the UV titration spectrum of TPE-DPY exhibited a persistent red-shift until stabilizing around 1.0 equivalent CB[8], and the related binding constant was obtained as $5.50 \times 10^6\,M^{-1}$ (Supplementary Fig. 20). Job's plot measured by UV-vis spectra confirmed a 1:1 stoichiometric ratio of TPE-DPY to CB[8] (Supplementary Fig. 21). Furthermore, two-dimensional rotating frame overhauser effect spectroscopy (2D ROESY) and two-dimensional diffusion-ordered spectroscopy (DOSY) were carried out to infer the binding mode. The correlation signals of proton $H_{b'}$ and $H_{g'}$ in TPE-DPY (Supplementary Fig. 22) manifested a deep encapsulation of bromophenylpyridine units by CB[8] cavity in a head-to-tail binding mode. The diffusion coefficients of guest molecule TPE-DPY ($D = 2.09 \times 10^{-10}\,m/s^2$) and assembly TPE-DPY/CB[8] ($D = 5.19 \times 10^{-11}\,m/s^2$) differed by an order of magnitude

(Supplementary Fig. 23), which verified the formation of n:n head-to-tail chain supramolecular pseudorotaxane for TPE-DPY/CB[8][46]. Correspondingly, transmission electron microscopy (TEM) and scanning electron microscopy (SEM) experiments revealed that the free guest molecule TPE-DPY presented ellipsoidal-shaped nanoparticles with sizes ranging from 50 to 90 nm (Fig. 2a, e, i), owing to the hydrophobic interaction and the stacking of tetraphenylethlene groups. In comparison, the TPE-DPY/CB[8] complex formed a nanorod with a length of approximately 500 nm (Fig. 2b, f, j), consistent with the head-to-tail chain pseudorotaxane assembly mode.

Unlike the complexation of TPE-DPY and CB[8], Job's plot determined by UV-vis spectra of TPE-DPY and CB[7] showed the inflection point at 0.2, implying a 1:4 stoichiometry ratio for TPE-DPY/CB[7] (Supplementary Fig. 24a). Nevertheless, no meaningful information was captured in $^1$H NMR titration experiments of TPE-DPY and CB[7] because of the strong proton peak passivation following the addition of CB[7] (Supplementary Fig. 25). Therefore, TPE-PY and SP-PY, consisting of the same functional groups as TPE-DPY, was selected as a reference compound for controlled experiments. Combined the $^1$H NMR titration spectrum (Fig. 3a and Supplementary Fig. 26) and two-dimensional correlation spectroscopy (2D COSY) (Supplementary Figs. 12 and 27), we found that upon the increasing amount of CB[7] from 0-1 equivalent, an apparent high-field shift of $H_a$ and $H_b$ on the bromophenylpyridine group was observed resulting from the shielding effect, while the protons $H_c$, $H_d$ and $H_e$ shifted download indicating that ethylene pyridine moiety was located outside of CB[7]. As the addition of CB[7] excessed 1 equivalent, $H_a$ and $H_b$ underwent a slight shift towards the low field concurrently accompanied by an up-field shift of $H_c$ and $H_d$. It indicates that CB[7] preferentially binds to the

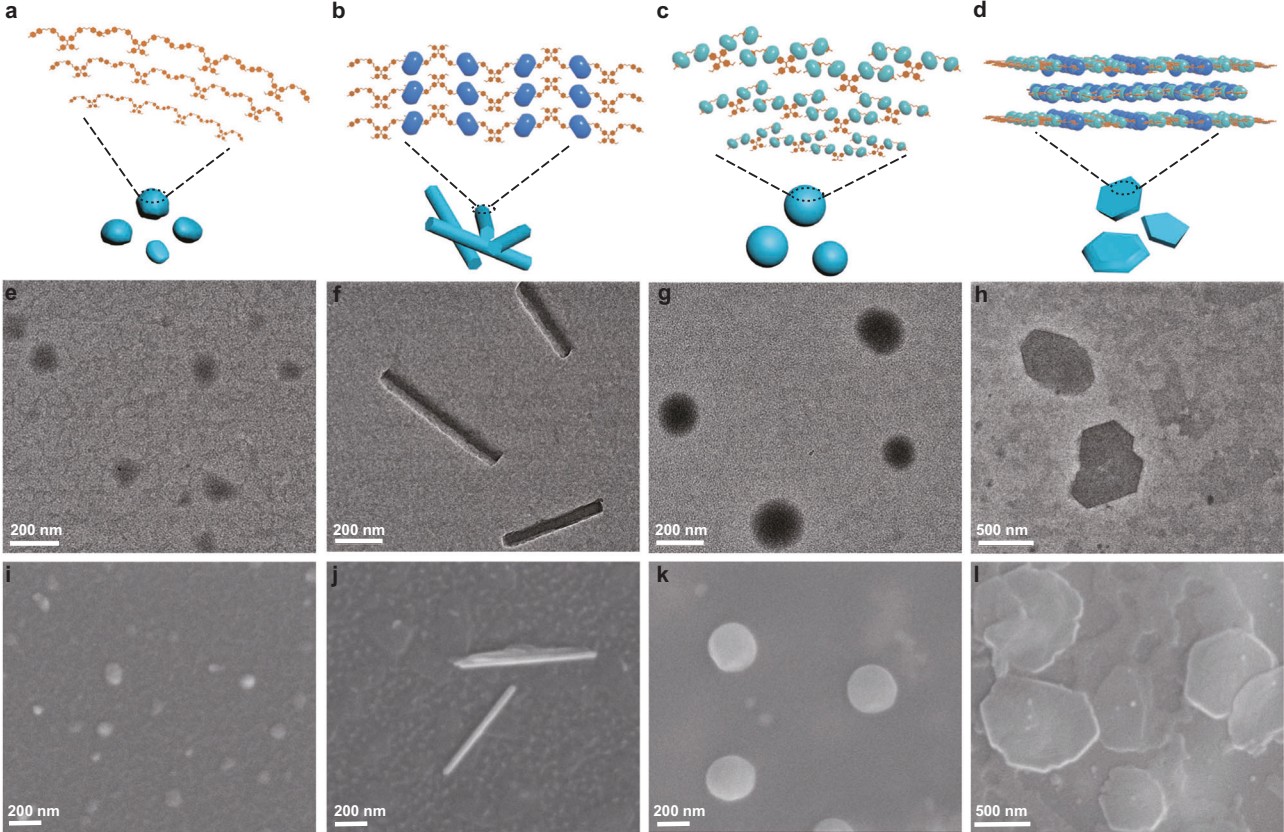

**Fig. 2 | Topological morphology characterization of TPE-DPY and the assemblies. a–d** Three-dimensional models of assembly structures. **e–h** TEM images of TPE-DPY ([TPE-DPY] = $1 \times 10^{-5}$ M), TPE-DPY/CB[8] ([TPE-DPY] = $1 \times 10^{-5}$ M, [CB[8]] = $1 \times 10^{-5}$ M), TPE-DPY/4CB[7] ([TPE-DPY] = $5 \times 10^{-6}$ M, [CB[7]] = $2 \times 10^{-5}$ M), TPE-DPY/CB[7]/CB[8]. ([TPE-DPY] = $1 \times 10^{-5}$ M, [CB[7]] = $2 \times 10^{-5}$ M, [CB[8]] = $1 \times 10^{-5}$ M) (from left to right). **i–l** SEM images of TPE-DPY, TPE-DPY/CB[8], TPE-DPY/4CB[7] and TPE-DPY/CB[7]/CB[8]. Each experiment was repeated three times independently with similar results.

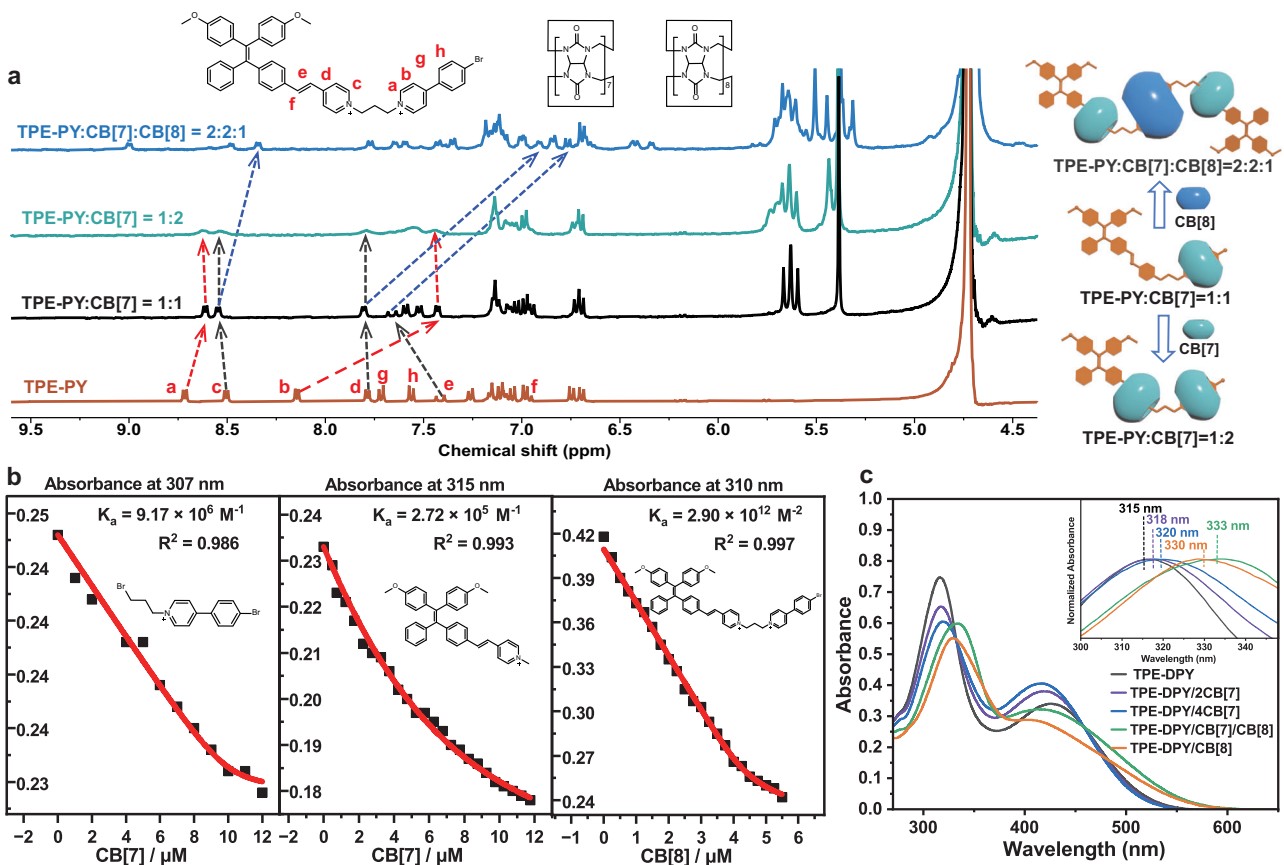

**Fig. 3 | Characterization of binding behavior between TPE-PY and CB[7]/CB[8].**
**a** $^1$H NMR spectra (400 MHz, D$_2$O with 10% DMSO-$d_6$, 298 K) of TPE-PY (red), TPE-PY:CB[7] = 1:1 (black), TPE-PY:CB[7] = 1:2 (green), TPE-PY:CB[7]:CB[8] = 2:2:1 (blue); **b** From left to right, the binding constants of PY-1 and TPE-2 with the addition of CB[7], and TPE-PY with the addition of CB[8]. **c** UV-vis absorption of TPE-DPY ([TPE-DPY] = 1 × 10$^{-5}$ M), TPE-DPY/2CB[7] ([TPE-DPY] = 1 × 10$^{-5}$ M, [CB[7]] = 2 × 10$^{-5}$ M), TPE-DPY/4CB[7] ([TPE-DPY] = 1 × 10$^{-5}$ M, [CB[7]] = 4 × 10$^{-5}$ M), TPE-DPY/CB[8] ([TPE-DPY] = 1 × 10$^{-5}$ M, [CB[8]] = 1 × 10$^{-5}$ M), TPE-DPY/CB[7]/CB[8]. ([TPE-DPY] = 1 × 10$^{-5}$ M, [CB[7]] = 2 × 10$^{-5}$ M, [CB[8]] = 1 × 10$^{-5}$ M).

bromophenylpyridine section at a low concentration and then assembles with the ethylene pyridine unit at a high concentration. The binding constants obtained through UV titration experiments give more proof for the above results, in which the binding constants of PY-1/CB[7] were brought to be 9.17 × 10$^6$ M$^{-1}$ higher than 2.72 × 10$^5$ M$^{-1}$ of TPE-2/CB[7] (Fig. 3b). Job's plot demonstrated a 1:2 stoichiometric ratio of TPE-PY to CB[7] (Supplementary Fig. 24b), further confirming the 1:4 binding mode between TPE-DPY and CB[7]. For SP-PY, the protons on ethylene pyridine and bromophenylpyridine units all showed obvious shifts to the high field with the addition of CB[7] and the cross-peaks signals between the protons of vinylpyridine, phenylpyridine and CB[7] were captured (Supplementary Fig. 28) providing another stronger support data for TPE-DPY/CB[7] binding mode. Moreover, $^1$H NMR titration and 2D NOESY spectrum of CB[7] and TPE-1 were performed to explore the effect of steric hindrance of neighboring ethylene pyridine units on binding mode. As shown in Supplementary Fig. 29, the protons on ethylene pyridine moiety (H$_{1-3}$) in reference molecule TPE-1 presented an apparent high-field shift upon the addition of CB[7] rely on the shielding effect, while protons on tetraphenylethylene units (H$_{5-8}$) shifted to low-field. The 2D NOESY spectrum of TPE-1/CB[7] presented the correlation signal between protons on pyridine units and CB[7] (Supplementary Fig. 30). The two-step binding constants of TPE-1 with CB[7] measured by UV-vis absorption titration are $K_1$ = 3.50 × 10$^5$ M$^{-1}$ and $K_2$ = 1.20 × 10$^5$ M$^{-1}$ (Supplementary Fig. 31a) and Job's plot showed a stoichiometric ratio of 1:2 stoichiometric ratio of TPE-1 to CB[7] (Supplementary Fig. 31b), revealing that CB[7] can effectively bond with neighboring vinylidene pyridine units. The assembly of TPE-DPY/CB[7] was visible in the TEM

and SEM images as nanospheres with the increased particle size caused by the rising CB[7] concentration. As shown in Fig. 2 and supplementary Fig. 32, a nanosphere measuring roughly 100 nm in diameter was produced by adding 2 equivalents of CB[7] and a large size nanosphere of about 200 nm developed as the amount of CB[7] increased to 4 equivalents, owing to the binding effect of CB[7] for ethylene pyridine portions and bromophenylpyridine groups, which increased the rigidity of supramolecular assembly and the space for stacking arrangement (Fig. 2c, g, k).

Interestingly, entirely varying from the binary assembly of TPE-DPY/CB[7] and TPE-DPY/CB[8], the morphology of TPE-DPY/CB[7]/CB[8] co-assembly exhibited a polyhedral nanoplate with distinct edges and corners in TEM images (Fig. 2d, h). Under a scanning electron microscope, it was evident that the three-dimensional nanoplates were formed by hierarchical self-assembly (Fig. 2l and Supplementary Fig. 32c), which resulted from the side-by-side and layer-by-layer stacking of TPE-DPY/CB[7]/CB[8] supramolecular assemblies with a sizeable rigid core and soft chains. The analysis of the co-assembly of the reference molecule TPE-PY with CB[7] and CB[8] allowed us to infer the binding mode of the ternary assembly TPE-DPY/CB[7]/CB[8]. Specifically, $^1$H NMR titration experiment showed that on the basis of TPE-PY/CB[7] with a stoichiometric ratio of 1:1 where CB[7] bound to the phenylpyridine unit, vinylpyridine protons (H$_c$, H$_d$, and H$_e$) exerted an high-field shift upon increase CB[8] concentration from 0 to 0.5 equivalent, indicating the tight encapsulation of ethylene pyridine moiety by CB[7] that moved from the phenylpyridine unit (Fig. 3a and Supplementary Figs. 33 and 34). Moreover, in the 2D NOESY spectrum of the TPE-PY/CB[7]/CB[8] assembly (Supplementary Fig. 35), we could

easily locate the cross peaks between the protons of vinyl functional groups and CB[7]. The 2:1 stoichiometry ratio obtained from Job's plot (Supplementary Fig. 36) and the strong binding constant of $2.90 \times 10^{12} \, \text{M}^{-2}$ for TPE-PY/CB[8] (Fig. 3b) provided further support for the aforementioned findings. These findings suggested that the vinylpyridine and phenyl pyridines moieties of TPE-PY were included in the cavities of CB[7] and CB[8], respectively, ultimately generating a ternary supramolecular assembly with a stoichiometric ratio of TPE-PY:CB[7]:CB[8] = 2:2:1. Thus, we deduced that the co-assembly of TPE-DPY/CB[7]/CB[8] went through a similar assembly process to form a linear supramolecular aggregate with a stoichiometric ratio of 1:2:1. From the above experimental results, it can be seen that CB[n] ($n$ = 7, 8) possesses a different binding affinity to TPE-DPY, leading to a diverse topological morphology for the supramolecular assembly. Profited by the host-guest complexation, hydrophobic interaction, and π – π stacking interactions, the binary assembly of TPE-DPY/CB[7] presents spherical nanoparticles with adjustable dimensions, and CB[8] with a larger hydrophobic cavity binds with TPE-DPY to form a n:n rod-shaped pseudorotaxane. The ternary co-assembly TPE-DPY/CB[7]/CB[8] has a more robust rigid structure than TPE-DPY/CB[7] and TPE-DPY/CB[8], resulting in a linear self-assembly stacked multi-layered three-dimensional nanoplates.

## Luminescence properties of supramolecular assemblies

Subsequently, the configuration-confined photophysical properties of the assembly of TPE-DPY and CB[n] ($n$ = 7, 8) were explored. With the binary assembly of CB[7] or CB[8], the absorption peak of TPE-DPY redshifted from 315 nm to 320 nm and 330 nm, respectively, and a comparable red shift by 18 nm occurred in the co-assembly of CB[7] and CB[8] (Fig. 3c). For the photoluminescence spectra shown in Fig. 4a, b, the guest molecule TPE-DPY exhibited a fluorescence emission at 390 nm with an excitation of 315 nm, and no phosphorescent signal was captured in the delayed spectrum. The assembly of TPE-DPY/2CB[7], TPE-DPY/4CB[7], TPE-DPY/CB[8], and TPE-DPY/CB[7]/CB[8] displayed distinct emission peaks around 530 nm as compared to the free TPE-DPY (Fig. 4a). Differing from the steady-state PL spectrum spectra, the delay spectra of these assemblies showed a major emission peak near 530 nm, illustrating its long-lived feature, which was further proved by their microsecond lifetime obtained by the time decay curve measurement (Fig. 4b, c). Notably, in contrast to the binary assembly TPE-DPY/2CB[7], TPE-DPY/4CB[7], and TPE-DPY/CB[8], the ternary assembly TPE-DPY/CB[7]/CB[8] had a stronger luminous intensity with extending the lifetime from 29.09 µs to 80.64 µs (Fig. 4c), because of the synergistic confinement effect of CB[7] and CB[8] on guest molecules and the supramolecular nanostructure formed by the linear rigid assembly layer by layer enabling valid shielding effect on the quencher. Additionally, after the injection of Ar, the lifetime of TPE-DPY/CB[7]/CB[8] aqueous solution at 540 nm was significantly increased from 80.64 µs to 122.58 µs with phosphorescence emission intensity increased by 2.5 times due to the avoidance of the triplet electron quenching caused by oxygen (Supplementary Fig. 37a, b). The temperature-dependent delayed spectrum presented a thermally quenched behavior at 540 nm, further confirming the phosphorescence properties (Supplementary Fig. 37c). The above experiment results demonstrated that the macrocyclic confinement can effectively induce a phosphorescence emission, and the topological morphology of supramolecule assembly can be regulated by adjusting the ratios of CB[7] and CB[8], presenting different photophysical properties.

## Characterization and optical properties of cascade assembly

On the basis of the binary supramolecular assembly, the multivalent cascade assembly has evolved into an effective method to improve phosphorescence performance[47]. Herein, HACD as a polysaccharide targeting agent has been introduced into TPE-DPY/CB[7]/CB[8] to construct a secondary assembly, which resulted in an exchanged topology from hierarchical self-assembled nanoplates to spherical nanoparticles (Figs. 2d and 5a). Dynamic light scattering (DLS), TEM, and zeta potential experiments were carried out to explore the assembly behavior for TPE-DPY/CB[7]/CB[8]@HACD. DLS measurements suggested that TPE-DPY/CB[7]/CB[8]@HACD assembly had an average hydrodynamic diameter of 236 nm, which matched the size of nanospheres in the TEM image (Fig. 5a, b). Moreover, on the contrary of TPE-DPY and TPE-DPY/CB[7]/CB[8] that possessed a positive zeta potential at +1.74 and +1.80 mV, respectively, a negative potential value of TPE-DPY/CB[7]/CB[8]@HACD was obtained as -0.335 mV (Supplementary Fig. 38), revealing the successful construction of the multi-component assembly. Significantly, the cascade assembly of TPE-DPY/CB[7]/CB[8]@HACD not only changed the topological morphology but also achieved a HACD-mediated single-molecule PRET based on macrocyclic confinement. As shown in Fig. 5e, f and Supplementary Fig. 39, upon the addition of HACD, the assembly of TPE-DPY/CB[7]/CB[8] excited by 333 nm showed a weak phosphorescence at 530 nm with a lifetime of 69.83 µs, and a dominant emission band centered at 700 nm with a lifetime of 21.60 µs, ascribing to the delayed fluorescence of the methoxytetraphenyl-vinylpyridine part stimulated by PRET.

## Mechanism for the phosphorescence and PRET

To further explore the phosphorescence luminescence mechanism and PRET luminescence behavior, a series of control experiments, density functional theory (DFT), and time-dependent density functional theory (TDDFT) calculations have been performed. The reference molecule PY-1 displayed CB[8]-induced strong RTP emission around 520 nm with a lifetime of 388.20 µs (Supplementary Fig. 40), suggesting that the phosphorescence emission of the TPE-DPY assembly emanated from the phenylpyridine units. The theoretical calculations results showed that there are only three possible transitions ($S_1 \rightarrow T_{2, 3, 4}$) for PY-1 intersystem hybridization, whereas, for PY-1/CB[8] assembly, more triplet states were close to the $S_1$ and the number of energy transfer channels was significantly increased ($S_1 \rightarrow T_{3, 4, 5, 6, 7, 8, 9}$) indicating that the confinement effect of CB[8] in promoting the intersystem crossing (ISC) (Supplementary Fig. 41). In addition, CB[8] with a strong hydrophobic cavity can not only limit the movement of the phenylpyridine unit but also effectively reduce the influence of quenching agents such as water, thus inhibiting the non-radiative transition process. The steady-state PL spectrum of TPE-1/CB[7] excited by 450 nm presented an emission peak at 720 nm (Fig. 5d) with a nanosecond lifetime measured as 0.39 ns, revealing the pure fluorescence properties of methoxy tetraphenylvinylpyridine unit (Supplementary Fig. 42). Similarly, under the excitation of 450 nm, TPE-DPY, TPE-DPY/CB[7]/CB[8] and TPE-DPY/CB[7]/CB[8]@HACD showed a fluorescence emission of 720 nm, where the lifetime was measured as 0.71 ns, 0.74 ns and 0.85 ns, respectively. No emission signal was obtained in the delay spectrum, verifying the property of HACD-mediated NIR delayed fluorescence at 700 nm under 333 nm excitation (Supplementary Fig. 43). Furthermore, a large overlap was captured between the absorption spectra of TPE-1/CB[7] and the phosphorescence spectra of PY-1/CB[8] (Fig. 5d), which provided a prerequisite for the PRET derived from phenyl pyridines unit to methoxy-tetraphenylethylene portion within a single-molecule. It is worth noting that the PRET phenomenon was also observed in the HACD-assembly doping system with PY-1/CB[8] as the donor and TPE-1/CB[7] as the acceptor. Upon increasing the donor/acceptor ratio from 20:1 to 1:1, the phosphorescence emission of donor intensity at 520 nm was gradually quenched accompanied by a concomitant enhancement of NIR emission intensity at 700 nm. The phosphorescence lifetime at 520 nm decreased from 82.48 to 21.52 µs, implying the PRET process between PY-1/CB[8] to TPE-1/CB[7] (Supplementary Fig. 44). Theoretical calculations showed that the HOMO-LUMO orbital

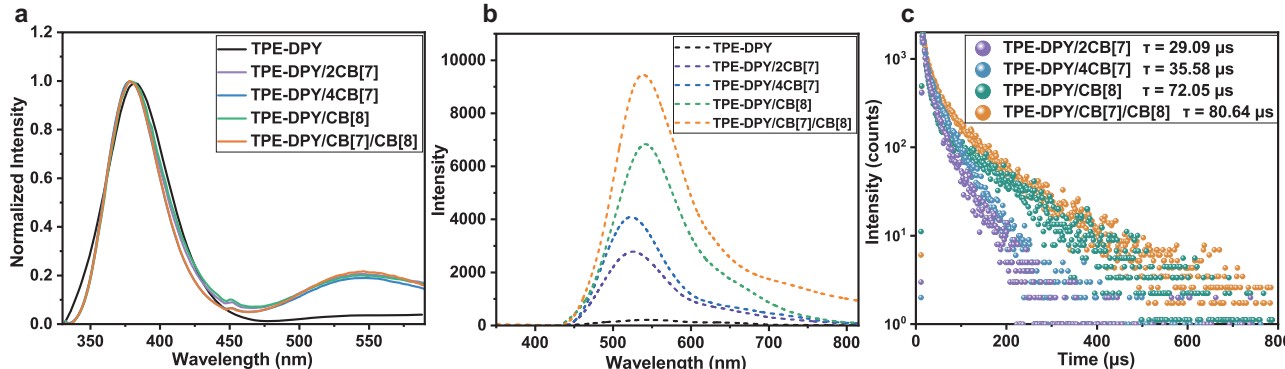

**Fig. 4 | Photophysical properties of TPE-DPY and the assemblies. a, b** The steady-state PL spectra and phosphorescence spectra of TPE-DPY ([TPE-DPY] = $1 \times 10^{-5}$ M, $\lambda_{ex}$ = 315 nm), TPE-DPY/2CB[7] ([TPE-DPY] = $1 \times 10^{-5}$ M, [CB[7]] = $2 \times 10^{-5}$ M, $\lambda_{ex}$ = 318 nm), TPE-DPY/4CB[7] ([TPE-DPY] = $1 \times 10^{-5}$ M, [CB[7]] = $4 \times 10^{-5}$ M, $\lambda_{ex}$ = 320 nm), TPE-DPY/CB[8] ([TPE-DPY] = $1 \times 10^{-5}$ M,

[CB[8]] = $1 \times 10^{-5}$ M, $\lambda_{ex}$ = 330 nm), TPE-DPY/CB[7]/CB[8] ([TPE-DPY] = $1 \times 10^{-5}$ M, [CB[7]] = $2 \times 10^{-5}$ M, [CB[8]] = $1 \times 10^{-5}$ M, $\lambda_{ex}$ = 333 nm). **c** Time-resolved decay curves of TPE-DPY/2CB[7], TPE-DPY/4CB[7], TPE-DPY/CB[8] and TPE-DPY/CB[7]/CB[8] in aqueous solution.

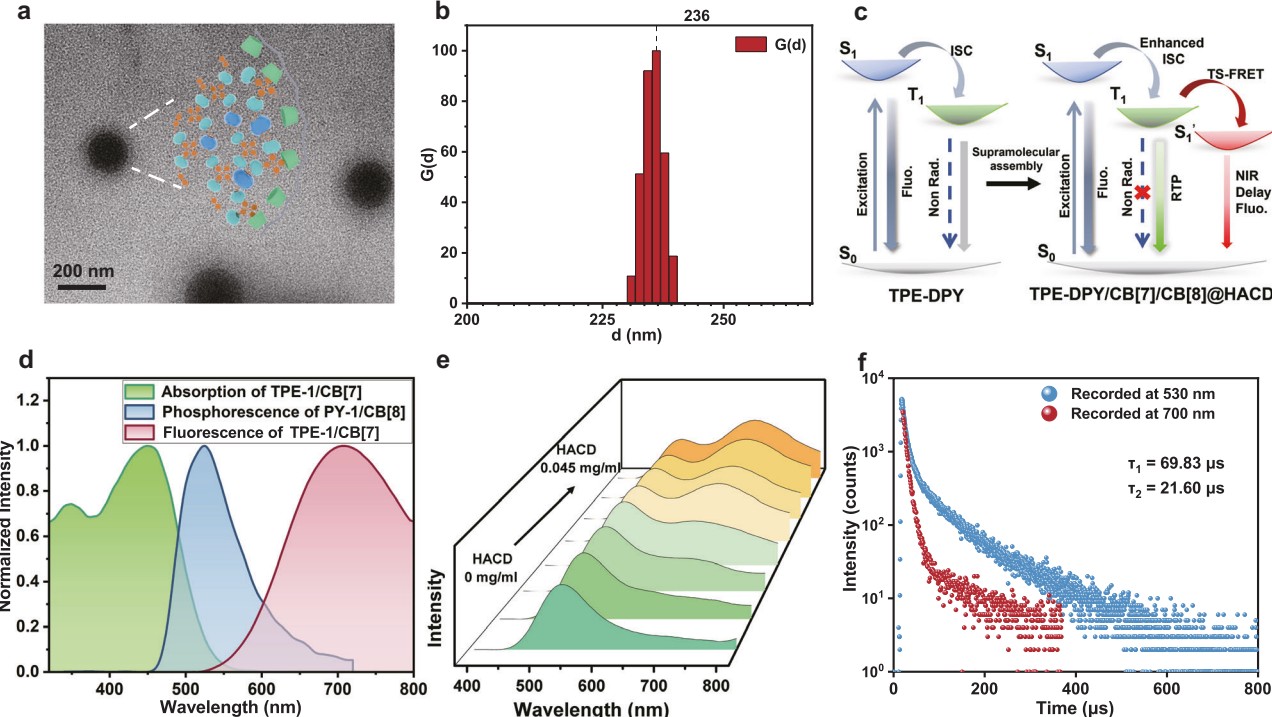

**Fig. 5 | HACD-mediated PRET process for the secondary assembly. a** TEM image of TPE-DPY/CB[7]/CB[8]@HACD (The experiment was repeated three times independently with similar results). **b** Size distribution of TPE-DPY/CB[7]/CB[8]@HACD determined by dynamic light scattering. **c** Schematic illustration of supramolecular assembly activated single-molecule PRET process. **d** Normalized phosphorescence emission spectrum of PY-1/CB[8] ([PY-1] = $2.5 \times 10^{-5}$ M, [CB[8]] = $1.25 \times 10^{-5}$ M), and

the excitation and emission spectra of TPE-1/CB[7] ([TPE-1] = $2.5 \times 10^{-5}$ M, [CB[7]] = $5 \times 10^{-5}$ M). **e** Phosphorescence spectra of TPE-DPY/CB[7]/CB[8] upon the addition of 0-0.045 mg/ml HACD ([TPE-DPY] = $2.5 \times 10^{-5}$ M, [CB[7]] = $5 \times 10^{-5}$ M, [CB[8]] = $2.5 \times 10^{-5}$ M). **f** The time-resolved decay curves of TPE-DPY/CB[7]/CB[8] @HACD aqueous solution record at 530 nm and 700 nm.

energy level of the triplet excited state of PY-1/CB[8] was close to that of the ground state of TPE-1/CB[7] (Supplementary Fig. 45), and the geometrically optimized molecular structure of TPE-DPY displayed that the central distance between the PY unit and the TPE moiety was ~ 1.191 nm (Supplementary Fig. 46), further confirming the possibility of resonance energy transfer process between PY/CB[8] to TPE/CB[7] moiety. In previous reports, β-CD has been shown to be able to effectively encapsulate methoxyphenyl functional groups to restrict the movement of guest molecules[22]. Therefore, the binding behavior between the assembly TPE-1/CB[7] and β-CD was investigated by ¹H NMR spectra. It was shown that upon the addition of β-CD, the protons on methoxyphenyl ($H_8$, $H_9$) in TPE-1 shifted slightly to high-field, while the protons in styryl pyridiniums remained unchanged (Supplementary Fig. 47), indicating the complexation of β-CD and methoxyphenyl unit, and the association constant of TPE-1/CB[7]/β-CD was determined to be 371.32 M⁻¹ (Supplementary Fig. 48). Moreover, the addition of HACD effectively enhanced the fluorescence emission of TPE-1/CB[7] with increased fluorescence lifetime (Supplementary Figs. 42 and 49), inferring that HACD could stabilize the singlet excitons of TPE-1/CB[7] which further enhance the possibility of PRET. These experimental results consistently indicated that the anion effect of HA and the encapsulation of β-CD to methoxy-tetraphenylethylene moiety

contributed to the reconstruction of topological morphology for TPE-DPY/CB[7]/CB[8], stabilizing the acceptor singlet excitons, and facilitating the single intramolecular PRET process leading to 700 nm NIR delayed fluorescence emission with large Stokes shift of 367 nm (Fig. 5c). Commonly, the phosphorescence spectra of the assembly TPE-DPY/CB[8]@HACD also exhibited NIR emission peaks at 700 nm, implying the universality of HACD-mediated single-molecule PRET process (Supplementary Fig. 50).

## Cancer cell-targeted imaging

In order to explore the application of macrocyclic confinement and HACD-mediated single-molecule PRET system, cell imaging experiments were constructed. First, Human cervical carcinoma cells (Hela cells) and human embryonic kidney cells (293T cells) were treated with TPE-DPY/CB[7]/CB[8]@HACD for 12 h, respectively, and then incubated with Hoechst and Mito-Tracker Green for localization experiment. The confocal laser scanning microscopy (CLSM) experiments were performed to investigate the intracellular NIR emission signals. As shown in Fig. 6a, c, Hela cells exhibited a bright NIR luminescence in a red channel (650–750 nm), whereas almost no red emission signal was found for normal 293T cells. These imaging results implied that TPE-DPY/CB[7]/CB[8]@HACD was preferentially internalized by cancer cells rather than normal cells, which may be caused by the overexpressed HA receptors for cancer cells. Furthermore, colocalization analysis demonstrated that the NIR luminescence signal overlapped well with the green signal of Mito Tracker, corresponding to the yellow region in the merged image (Fig. 6b). It revealed the ability of TPE-DPY/CB[7]/CB[8]@HACD for targeted mitochondria imaging in cancer

cells, and the high Pearson correlation provided strong evidence for this result (Supplementary Fig. 51). Finally, CCK-8 assays were conducted to evaluate cytotoxicity experiments on the above two cells, and the high survival rate indicated low cytotoxicity of the assembly TPE-DPY/CB[7]/CB[8]@HACD (Supplementary Fig. 52).

## Discussion

In summary, a single-molecule PRET is activated by the macrocyclic confinement of CB[n] (n = 7, 8) and the secondary assembly of HACD to achieve NIR-delayed fluorescence emission. Based on the different cavity sizes, the primary assembly of CB[7] and CB[8] to TPE-DPY presented a macrocyclic confinement-induced phosphorescence behavior accompanied by controllable topological morphologies, which realized a transformation from nanosphere, rod-shaped pseudorotaxane to hierarchical self-assembled nanoplate. Especially, the secondary assembly of HACD activated the single intramolecular PRET from the phenyl pyridines unit to the methoxy-tetraphenylethylene portion, generating an NIR delayed fluorescence emission at 700 nm. Different from the dye-doped PRET system, this macrocyclic-confinement polysaccharide activated single molecular PRET system displayed a large Stokes shift of 367 nm and was successfully applied in mitochondrial-targeted imaging for cancer cells, which provides an alternative approach for the construction and application of single-molecule PRET.

## Methods
### Materials
All research complied with the ethical regulations and was approved by the Nankai University committees. Except for additional stated, all

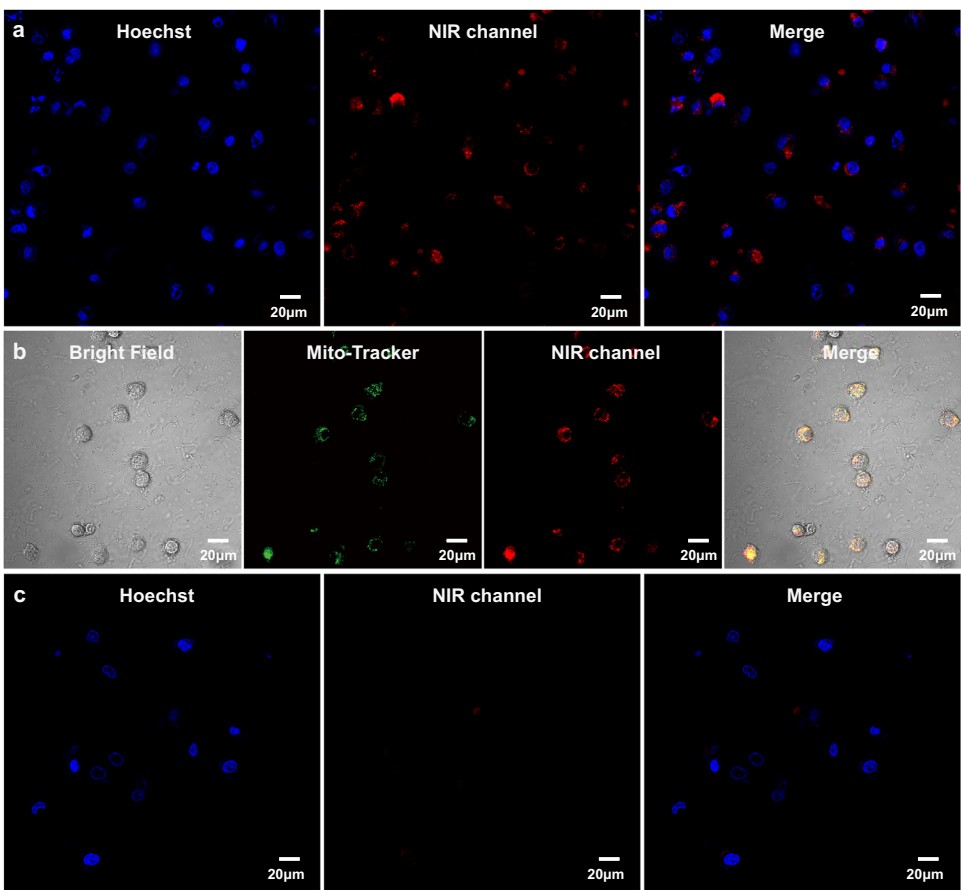

**Fig. 6 | Application of TPE-DPY/CB[7]/CB[8]@HACD in targeted imaging for cancer cells. a, b** Confocal microscopy images and merged images of Hela cells in the presence of TPE-DPY/CB[7]/CB[8]@HACD, Hoechst and Mito-Tracker Green. **c** Confocal microscopy images and merged images of 293T cells in the presence of TPE-DPY/CB[7]/CB[8]@HACD and Hoechst. ([TPE-DPY] = 1 × 10$^{-5}$ M, [CB[7]] = 2 × 10$^{-5}$ M, [CB[8]] = 1 × 10$^{-5}$ M, [HACD] = 0.018 mg/ml, Green channel: 450–550 nm, red channel: 650–800 nm, each experiment was repeated two times independently with similar results).

reagents and solvents were available from commercial sources and used directly without any purification. [1]H NMR and [13]C NMR spectrums were recorded through Bruker AV400. Two-dimensional NMR (COSY, DOSY, NOESY, and ROESY) spectra were measured on Bruker AVANCE III HD 400 spectrometer. High-resolution mass spectrometry (HR-MS) was recorded on a Q-TOF LC-MS in an Electrospray ionization (ESI) source. UV-vis absorption was kept details on Shimadzu UV-3600 spectrophotometer with a PTC-348WI temperature controller at 298 K. Photoluminescence (PL) spectrum and time-correlated decay profiles were documented on Edinburgh Instruments F900. The TEM experiment was carried out on FEI Tecnai G2 F20 under 200 KV. SEM was accomplished on FEI Apreo S LoVac scanning electronic microscope working at an accelerating voltage of 30 keV. The Zeta potentials were examined on Brookhaven ZETAPALS/BI-200SM at 298 K. Dynamic Light Scattering (DLS) was determined by using a laser lights-scattering spectrometer (BI-200SM) equipped with a digital correlator (Turbo Corr) at 635 nm at a scattering angle of 90°. Cell images were captured on Olympus FV1000 Laser scanning confocal microscope.

## Cytotoxicity experiments

The Hela (ATCC, CCL-2) cell line and 293T (ATCC, CRL-3216) cell line, were all gained from the Cell Resource Center of China Academy of Medical Science in Beijing. These cells were cultured in particular conditions with the addition of 10% FBS and 1% penicillin/streptomycin in the DMEM nutrient medium and humidified incubator with 5% $CO_2$ atmosphere at 37 °C. The Hela cells and 293T cells were incubated with TPE-DPY/CB[7]/CB[8]@HACD at distinct concentrations in 96-well plates for 24 hours. Then assayed for cell viability with the CCK-8 Kit according to the manufacturer's instructions. After adding CCK-8, cells were incubated for another 2 h. Finally, the absorbance at 450 nm was recorded by a microplate reader (Varioskan LUX). The cytotoxicity was presented as the relative percentage of the cell viability compared with the control group.

## Cell imaging experiments

The Hela cells and 293T cells were seeded in confocal petri dishes at a density of $3 \times 10^4$ cells per well in 1 mL of complete culture medium, respectively, for 24 h before treatment. Then, the well-cultured cells were incubated with TPE-DPY/CB[7]/CB[8]@HACD (10 μM) for a further 12 h. The cells were stained with Hoechst (1 nM) and Mito-Tracker Green (1 nM) for half an hour, washed three times with PBS, and then added PBS (1 ml) to observe by microscope.

## Computational methods

Geometry optimization of TPE-DPY, PY-1, supramolecular assembly PY-1/CB[8] and TPE-1/CB[7] were performed in Gaussian 16, Revision C.02 program using M06-2X functional with D3 dispersion correction and 6-31 G(d) basis set with SMD model (water as solvent). Further information is provided in the Supplementary Information and the atomic coordinates of the optimized computational models are listed in Supplementary Data 1.

## Synthesis of compound TPE-DPY

Under $N_2$ protection, 4,4'-[[2,2-bis(4-methoxyphenyl)ethenylidene]bis(4,1-phenylene-2,1-ethenediyl)]pyridine (54 mg, 0.09 mmol) and PY-1 (94.5 mg, 0.22 mmol) were dissolved in $CH_3CN$ (5 ml). The reaction mixture was heated to 85 °C for 36 hours. After the reaction, $CH_3CN$ was evaporated, and acetone was used for ultrasound cleaning. Then, the mixture was filtered and washed with acetone twice. Finally, the crude powder was purified by heat filtration and recrystallization to give an orange solid (18 mg, yield: 13.7%). [1]H NMR (400 MHz, Methanol-$d_4$) δ 9.07 (d, $J = 6.7$ Hz, 4H), 8.86 (d, $J = 6.5$ Hz, 4H), 8.49 (d, $J = 6.5$ Hz, 4H), 8.20 (d, $J = 6.6$ Hz, 4H), 7.97 (d, $J = 8.5$ Hz, 4H), 7.91 (d, $J = 16.2$ Hz, 2H), 7.84 (d, $J = 8.5$ Hz, 4H), 7.56 (d, $J = 8.1$ Hz, 4H), 7.39 (d, $J = 16.2$ Hz, 2H), 7.15 (d, $J = 8.2$ Hz, 4H), 6.98 (d, $J = 8.6$ Hz, 4H), 6.72 (d, $J = 8.7$ Hz, 4H), 4.84 (s, 4H), 4.76 (t, $J = 7.5$ Hz, 4H), 3.75 (s, 6H), 2.84 – 2.78 (m, 4H).; [13]C

NMR (101 MHz, Methanol-$d_4$) δ 158.97, 155.76, 154.58, 146.95, 144.83, 143.98, 141.73, 135.52, 133.19, 132.80, 132.42, 131.95, 129.61, 127.80, 127.00, 124.92, 124.01, 122.16, 112.94, 57.27, 56.79, 54.26, 31.99.; HRMS (ESI) m/z for $C_{70}H_{62}Br_6N_4O_2$ calcd. [M-4Br]$^{4+}$ 287.5799, found: 287.5803.

## Reporting summary

Further information on research design is available in the Nature Portfolio Reporting Summary linked to this article.

## Data availability

The authors declare that the data supporting the findings of this study are available within the paper and its supplementary information files. Extra data are available from the corresponding author upon request. Source data are provided with this paper.

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

## Acknowledgements

We thank the National Nature Science Foundation of China (NNSFC, Grant Nos. 22131008 to Y.L. and 22271165 to H.-Y.Z.) for financial support.

## Author contributions

X.Z. and X.B. contributed equally to this work and designed and performed all experiments. X.Z. wrote the manuscript. H.-Y.Z. and L.-H.W. gave valuable advice on data analysis and revision. F.S. and X.X. performed the theoretical calculations. Y.L. supervised the work and edited the manuscript. All authors analyzed and discussed the results and reviewed the manuscript.

## Competing interests

The authors declare no competing interests.
