## [Peer Review File · Nature Communications]

Supramolecular Assembly Activated Single-Molecule
Phosphorescence Resonance Energy Transfer for Near-
Infrared Targeted Cell ImagingREVIEWER COMMENTS

Reviewer #1 (Remarks to the Author):

This study involved the synthesis of a novel supramolecular assembly, consisting of alkyl-bridged methoxy-tetraphenylethylene-phenylpyridines derivative (TPE-DPY), cucurbit[n]uril, and β -CD grafted hyaluronic acid (HACD), leading to the formation of a single-molecule phosphorescence resonance energy transfer (PRET) which was successfully applied in tumor-targeted near-infrared imaging. Cucurbit[n]uril (CB[n]) is employed for inducing the phosphorescence of phenyl pyridines unit from TPE-DPY, while based on the different binding affinities, the primary assembly TPE-DPY/CB[n] presented controllable topological morphologies, realizing a transformation from nanosphere, rod-shaped pseudorotaxane to hierarchical self-assembled nanoplate. The introduction of HACD further changes the topological morphology from nanoplate to nanosphere and activates single intramolecular PRET from the phenyl pyridines unit to the methoxy-tetraphenylethylene portion, ultimately enabling near-infrared targeted imaging of cancer cells. Different from the dye-doped PRET system, this work releases a new concept that the macrocyclic confinement of CB[n] ($n = 7, 8$) and the cascade assembly of HACD could cooperatively activate the single molecule PRET to achieve a large Stokes shift. And, this work carries great significance in guiding the future development of supramolecular assembly and room-temperature phosphorescence. The present findings are highly valuable and, following some suggested revisions, so it is recommended for publication in Nature Communications. The specific comments for minor revision are as follows.

1. On page 2, ‘... achieving tunable phosphorescence emission, especially in the near-infrared (NIR) region still faces great challenges owing to the limitation of the energy gap law.²⁸’ The citation of reference 28 seems inappropriate, and the authors need to supplement and cite appropriate references.
2. In Figure 2d, the legends of the inset figure of normalized absorption spectra seem to be not clear. Please check and revise it.
3. The scale bars in Figure 3 and Figure S29 should be redrawn for clearer exhibition.
4. On page 17 of the manuscript, according to the supporting information, the value of C70H62Br6N4O2 [M-4Br]₄⁺ should be 287.5803 not 287.0806. Please check and revise it.
5. Ensure consistency in font size for the manuscript, for example, on page 6, line 127, ‘Supplementary’.
6. Phosphorescence resonance energy transfer and room temperature phosphorescence materials are hot topics in chemistry and materials. To arouse a broad interest from readership in this field, some strongly related works on phosphorescence energy transfer (Nat. Commun. 2023, 14, 1654; Adv. Funct. Mater. 2023, 33, 2300735) and room temperature phosphorescence (CCS Chemistry, 2023, 5, 2866; Angew. Chem. Int. Ed. 2023, 62, e202309913) could be added as references.
7. Page 8, line 159. ‘TPE- PY:CB[7]:CB[8]’ should be modified to ‘TPE-PY:CB[7]:CB[8]’. Please double check and revise it.
8. Page 12, line 245. ‘(Supplementary Fig. S38)’ should be ‘(Supplementary Fig. 38)’.

Reviewer #2 (Remarks to the Author):

The authors present a supramolecular assembly activated single-molecule phosphorescence resonance energy transfer (PRET) system which is further useful in mitochondrial targeted imaging. Differing from previous PRET systems by dye-dope or noncovalently assembly of Donor and Acceptor, in this work a single molecule TPE-DPY (D and A covalently linked) is designed and its PRET feature is activated by the secondary supramolecular assembly. No doubt this interesting work provides a new strategy for the construction and application of PRET system based on single-molecule. I thus recommend the publication of this work in Nat. Commun., after few points are addressed.

1. Although the PRET system is based on the single molecule TPE-DPY, the PRET nature of this molecule does not display until it is activated by the secondary supramolecular assembly. The title “Single-Molecule Phosphorescence Resonance Energy Transfer for NIR Targeted Cell Imaging” overemphasizes “Single-Molecule” PRET and does not fit the work. More specific title, such as “supramolecular assembly activated single-molecule phosphorescence resonance ...”, is suggested.
2. Although TPE-PY is selected as a reference compound for controlled experiments, its binding modes with CB7 or CB7/CB8 may not exactly fit the binding of TPE-DPY with CB7 or CB7/CB8, since the steric hindrance of CB7 should also be considered when assuming two CB7 binding on the two neighboring ethylene pyridine units. Since the following results (TPE-DPY/CB7/CB8 co-assembly) are based on the formation of TPE-PY-CB7 inclusion complex with binding ratio at 4:1, the authors are strongly suggested to provide more solid evidence such as ESI-MS to prove this 4:1 binding ratio. Similarly, how do the authors exclude the presence of TPE-DPY/CB[8] and TPE-DPY/CB[7] in the proposed co-assembly of TPE-DPY/CB[7]/CB[8]? The photophysical properties and morphology could be affected if the ternary co-assembly TPE-DPY/CB[7]/CB[8] is not exclusively formed.
3. The authors mentioned that “the protons on methoxyphenyl (H1, H2) in TPE-1 shifted slightly to low-field, while the protons in styryl pyridiniums remained unchanged (Supplementary Fig. S38), indicating the complexation of β -CD and methoxyphenyl unit”. But in Fig S38, all proton signals at low field, including protons in styryl pyridinium showed downfield shifts as the same as protons in styryl pyridinium. The Binding mode and K_a should be reevaluated.
4. Normally, the figures/schemes are numbered according to the text sequence. Based on that, Fig 3 and Fig 2 should be re-numbered reversely and removed to the corresponding word description. Fig S15 and Fig S23 should be removed to the compounds characterization part since they are COSY spectra of guest molecule rather than host-guest complexes.

Reviewer #3 (Remarks to the Author):

This manuscript by Zhou et al. reports the construction of a single-molecule phosphorescence resonance energy transfer (PRET) system with near-infrared (NIR) emission by using alkyl-bridged

methoxy-tetraphenylethylene phenylpyridines derivative (TPE-DPY), cucurbit[n]uril (CB[n], n=7,8), and β -cyclodextrin modified hyaluronic acid (HACD). The authors have shown the RTP energy transfer in a single molecule containing both donor (phenyl pyridines unit) and acceptor (methoxy-tetraphenylethylene portion) through supramolecular confinement and generated an NIR delayed fluorescence emission at 700 nm which has been applied for mitochondrial-targeted imaging for cancer cells. I find that this work is a follow up/extension of their recently reported work (Adv. Mater. 2022, 34, 2203534), in which the authors have reported a supramolecular confinement RTP-harvesting assembly $G \subset CB[7]@HACD$. This system shows efficient energy transfer to externally doped Nile blue or tetrakis(4-sulfophenyl)porphyrin dye, accompanied by a long-lived NIR-emission (680 and 710 nm) which has also been applied for targeted NIR imaging of living tumor cells. On the overall examination and based on the points mentioned below, I don't see any novelty and clarity in this work and hence, I do not recommend the publication of this manuscript in the high impact Nature Communication journal.

Points of concern:

1. There are several literatures regarding the excimer formation in the presence of CB8. Generally, the excimers have the emission band in the longer wavelength region w.r.t the monomer emission as well as the lifetime increases to μ s. TPE-DPY molecule has very low fluorescence yield. By forming self-assembly with CB8, the fluorescence intensity increases and the peak position at 540 nm matches well with the emission maxima of the aggregated TPE reported in the literature (J. Am. Chem. Soc. 2011, 133, 50, 20126). In the presence of CB8, TPE-DPY undergoes dimerization and shows fluorescence. The authors may have a look at the data from the above point as well.
2. The interpretation for phosphorescence spectrum of TPE-DPY in the presence of CB8 is mainly based on the qualitative data. The authors need to provide theoretical data to support their claim.
3. The authors state that "Due to the restriction of phenyl-pyridine unit motion by cucurbituril hydrophobic cavities through host-guest complexation, the binary assembly of CB[7] or CB[8] to TPE-DPY all induced a distinct intense phosphorescent emission around 530 nm" Mechanistically how does the restriction of phenyl-pyridine unit motion facilitates triplet population? If this is correct, restriction induced by any other means (say by rigid medium) also should do the same!
5. Why no role of TPE is discussed in the formation of assembly? Only place it was discussed is for the HACD interaction at the methoxy group. But what is the electronic mechanism (other than topological change) by which NIR delayed fluorescence is induced?
6. It was stated that "...after the injection of Ar, the lifetime of TPE -DPY/CB[7]/CB[8] aqueous solution at 540 nm was significantly increased from 59.36 μ s to 129.97 μ s (Supplementary Fig .30) due to the avoidance of the triplet electron quenching caused by oxygen, further confirming the phosphorescence properties of emission peak at 540 nm" My observation on the trace of Fig.30 says that the decay traces carry two lifetimes (one fast and other slow) and only change is in their relative amplitude contribution and the lifetime values may remain the same. Why a proper analysis is missed?

7. The authors have qualitatively described and schematically shown the interactions and assembly formation without adequate quantitative data. How with the presence of such bulky macrocyclic groups the assembly formation is feasible?

8. How do the authors justify the efficient energy transfer from one part of the complex to another part and also with HACD through space? It should have been adequately supported by the theoretical modelling justifying the placing of energy levels and their overlap function etc.

9. I agree that the provided NMR data indicate the interactions, but not the mechanism of triplet to singlet energy transfer.

Reply comments

Reply to reviewer 1.

Reviewer #1 (Remarks to the Author):

This study involved the synthesis of a novel supramolecular assembly, consisting of alkyl-bridged methoxy-tetraphenylethylene-phenylpyridines derivative (TPE-DPY), cucurbit[n]uril, and β -CD grafted hyaluronic acid (HACD), leading to the formation of a single-molecule phosphorescence resonance energy transfer (PRET) which was successfully applied in tumor-targeted near-infrared imaging. Cucurbit[n]uril (CB[n]) is employed for inducing the phosphorescence of phenyl pyridines unit from TPE-DPY, while based on the different binding affinities, the primary assembly TPE-DPY/CB[n] presented controllable topological morphologies, realizing a transformation from nanosphere, rod-shaped pseudorotaxane to hierarchical self-assembled nanoplate. The introduction of HACD further changes the topological morphology from nanoplate to nanosphere and activates single intramolecular PRET from the phenyl pyridines unit to the methoxy-tetraphenylethylene portion, ultimately enabling near-infrared targeted imaging of cancer cells. Different from the dye-doped PRET system, this work releases a new concept that the macrocyclic confinement of CB[n] ($n = 7, 8$) and the cascade assembly of HACD could cooperatively activate the single molecule PRET to achieve a large Stokes shift. And, this work carries great significance in guiding the future development of supramolecular assembly and room-temperature phosphorescence. The present findings are highly valuable and, following some suggested revisions, so it is recommended for publication in Nature Communications. The specific comments for minor revision are as follows.

Reply: We are grateful for the reviewer's positive comments. A point-by-point reply is attached below.

1. On page 2, '... achieving tunable phosphorescence emission, especially in the near-infrared (NIR) region still faces great challenges owing to the limitation of the energy gap law.²⁸' The citation of reference 28 seems inappropriate, and the authors need to supplement and cite appropriate references.

Reply: We appreciate the reviewer for the good advice. According to the comment, we have replaced the original reference with a more appropriate reference in the revised manuscript. **(Page 21, Ref 31)**

2. In Figure 2d, the legends of the inset figure of normalized absorption spectra seem to be not clear. Please check and revise it.

Reply: We appreciate the reviewer's comment. According to the reviewer's advice, we have improved the clarity of the original inset figure of normalized absorption spectra. **(Page 9)**

3. The scale bars in Figure 3 and Figure S29 should be redrawn for clearer exhibition.

Reply: We appreciate the reviewer's advice. We have re-labeled the scale bars in TEM and SEM images in the revised manuscript. **(Page 6; SI, Page 21)**

4. On page 17 of the manuscript, according to the supporting information, the value of $C_{70}H_{62}Br_6N_4O_2 [M-4Br]_{4+}$ should be 287.5803 not 287.0806. Please check and revise it.

Reply: We appreciate the reviewer's advice. According to the reviewer's advice, we have corrected the value of $C_{70}H_{62}Br_6N_4O_2$ to 287.5803. **(Page 18, line 372)**

5. Ensure consistency in font size for the manuscript, for example, on page 6, line 127, 'Supplementary'.

Reply: We appreciate the reviewer's comments. The font size has been consistent throughout the revised manuscript.

6. Phosphorescence resonance energy transfer and room temperature phosphorescence materials are hot topics in chemistry and materials. To arouse a broad interest from readership in this field, some strongly related works on phosphorescence energy transfer (Nat. Commun. 2023, 14, 1654; Adv. Funct. Mater. 2023, 33, 2300735) and room temperature phosphorescence (CCS Chemistry, 2023, 5, 2866; Angew. Chem. Int. Ed. 2023, 62, e202309913) could be added as references.

Reply: We appreciate this helpful advice. According to the reviewer's comments, we have added Refs. 28, 29, 34 and 37 in the revised manuscript. **(Pages 21, 22)**

7. Page 8, line 159. 'TPE- PY:CB[7]:CB[8]' should be modified to 'TPE- PY:CB[7]:CB[8]'. Please double-check and revise it.

Reply: We appreciate the reviewer's advice. We have double-checked and corrected 'TPE-PY:CB[7]:CB[8]' in the revised manuscript. **(Page 8)**

8. Page 12, line 245. '(Supplementary Fig. S38)' should be '(Supplementary Fig. 38)'.

Reply: We appreciate the reviewer's advice. We have changed the "Supplementary Fig. S38" to "Supplementary Fig. 38". **(Page 11)**

Reply to reviewer 2.

Reviewer #2 (Remarks to the Author):

The authors present a supramolecular assembly activated single-molecule phosphorescence resonance energy transfer (PRET) system which is further useful in mitochondrial targeted imaging. Differing from previous PRET systems by dye-dope or noncovalently assembly of Donor and Acceptor, in this work a single molecule TPE-DPY (D and A covalently linked) is designed and its PRET feature is activated by the secondary supramolecular assembly. No doubt this interesting work provides a new strategy for the construction and application of PRET system based on single-molecule. I thus recommend the publication of this work in Nat. Commun., after few points are addressed.

Reply: We are grateful for the reviewer's positive comments. A point-by-point reply is attached below.

1. Although the PRET system is based on the single molecule TPE-DPY, the PRET nature of this molecule does not display until it is activated by the secondary supramolecular assembly. The title "Single-Molecule Phosphorescence Resonance Energy Transfer for NIR Targeted Cell Imaging" overemphasizes "Single-Molecule" PRET and does not fit the work. More specific title, such as "supramolecular assembly activated single-molecule phosphorescence resonance ...", is suggested.

Reply: We appreciate the reviewer's comment. According to the reviewer's advice, we have changed the title to "Supramolecular assembly activated single-molecule phosphorescence resonance energy transfer for NIR targeted cell imaging" in the revised manuscript and supporting information.

2. Although TPE-PY is selected as a reference compound for controlled experiments, its binding modes with CB7 or CB7/CB8 may not exactly fit the binding of TPE-DPY with CB7 or CB7/CB8, since the steric hindrance of CB7 should also be considered when assuming two CB7 binding on the two neighboring ethylene pyridine units. Since the following results (TPE-DPY/CB7/CB8 co-assembly) are based on the formation of TPE-PY-CB7 inclusion complex with binding ratio at 4:1, the authors are strongly suggested to provide more solid evidence such as ESI-MS to prove this 4:1 binding

ratio. Similarly, how do the authors exclude the presence of TPE-DPY/CB[8] and TPE-DPY/CB[7] in the proposed co-assembly of TPE-DPY/CB[7]/CB[8]? The photophysical properties and morphology could be affected if the ternary co-assembly TPE-DPY/CB[7]/CB[8] is not exclusively formed.

Reply: We appreciate the reviewer's comment. According to the reviewer's comments, reference molecule TPE-1 with two neighboring ethylene pyridine units has been synthesized to study the steric hindrance of two CB[7]. In control experiments, ¹H NMR titration experiments, 2D COSY, 2D NOESY spectrum and Job's plot have been performed to confirm a stoichiometric ratio of 1:2 TPE-1 to CB[7] and binding mode of TPE-DPY/CB[7]. The more results and discussions have been added to the revised manuscript (**Page 7, lines 138-147; SI, Supplementary Figs. 27, 28, 29**). To provide more solid evidence for the 4:1 binding ratio of TPE-DPY/CB[7], the ESI-MS experiments of TPE-DPY/4CB[7] have been performed, unfortunately, due to the multi-charge nature of the assembly, no valid ESI-MS results have been obtained. Therefore, reference molecule SP-PY with alkyl-bridged ethylene pyridine and bromophenylpyridine units was synthesized to investigate the binding mode. ¹H NMR titration experiments, 2D COSY, 2D NOESY spectrum have been performed to prove that CB[7] binds to the arms at vinylpyridine and phenylpyridine unit, respectively, giving the binding mode of TPE-DPY/CB[7]. The relevant results and discussions have been added in the revised manuscript (**SI, pages 19-20, Supplementary Fig. 30**).

Supramolecular self-assembly is a dynamic equilibrium process. According to the different binding affinity of CB[7] ($K_s \sim 10^5 \text{M}^{-1}$) and CB[8] ($K_s \sim 10^6 \text{M}^{-1}$) to TPE-DPY and bonding site (**Fig. 3**), we first added CB[7] to the TPE-DPY solution to form a binary assembly TPE-DPY/2CB[7], then added CB[8] for secondary assembly finally obtaining a ternary assembly TPE-DPY/CB[7]/CB[8], which efficiently avoids the binary assemblies interference exhibiting effectively enhanced phosphorescence behavior (**Fig. 4b**). On the other hand, the TEM and SEM experiments clearly showed multi-layered three-dimensional nanoplates of TPE-DPY/CB[7]/CB[8], which are different from the nanoparticles and nanorods of binary assemblies for CB[7] and CB[8], respectively (**Fig. 2**). Further results and discussions are presented in the revised manuscript. (**Page**

8, paragraph 2)

3. The authors mentioned that “the protons on methoxyphenyl (H1, H2) in TPE-1 shifted slightly to low-field, while the protons in styryl pyridiniums remained unchanged (Supplementary Fig. S38), indicating the complexation of β -CD and methoxyphenyl unit”. But in Fig S38, all proton signals at low field, including protons in styryl pyridinium showed downfield shifts as the same as protons in styryl pyridinium. The Binding mode and K_a should be reevaluated.

Reply: We appreciate the reviewer’s comment. According to the reviewer’s advice, we have added the new experiments and reevaluated the binding mode and K_a . The detailed results and discussions have been added to the revised manuscript. **(Page 13, lines 272-276; SI, Supplementary Figs. 46, 47)**

4. Normally, the figures/schemes are numbered according to the text sequence. Based on that, Fig 3 and Fig 2 should be re-numbered reversely and removed to the corresponding word description. Fig S15 and Fig S23 should be removed to the compounds characterization part since they are COSY spectra of guest molecule rather than host-guest complexes.

Reply: We appreciate the reviewer’s comment. According to the reviewer’s advice, Fig 3 and Fig 2 have been re-numbered reversely and removed to the corresponding word description. Fig S15 and Fig S23 in the original manuscript have been removed to the compound’s characterization part. **(Pages 6 and 9; SI, pages 7 and 10)**

Reply to reviewer 3.

Reviewer #3 (Remarks to the Author):

This manuscript by Zhou et al. reports the construction of a single-molecule phosphorescence resonance energy transfer (PRET) system with near-infrared (NIR) emission by using alkyl-bridged methoxy-tetraphenylethylene phenylpyridines derivative (TPE-DPY), cucurbit[n]uril (CB[n], n=7,8), and β -cyclodextrin modified hyaluronic acid (HACD). The authors have shown the RTP energy transfer in a single molecule containing both donor (phenyl pyridines unit) and acceptor (methoxy-tetraphenylethylene portion) through supramolecular confinement and generated an NIR delayed fluorescence emission at 700 nm which has been applied for mitochondrial-targeted imaging for cancer cells. I find that this work is a follow up/extension of their recently reported work (Adv. Mater. 2022, 34, 2203534), in which the authors have reported a supramolecular confinement RTP-harvesting assembly $G\text{-CB}[7]\text{@HACD}$. This system shows efficient energy transfer to externally doped Nile blue or tetrakis(4-sulfophenyl)porphyrin dye, accompanied by a long-lived NIR-emission (680 and 710 nm) which has also been applied for targeted NIR imaging of living tumor cells. On the overall examination and based on the points mentioned below, I don't see any novelty and clarity in this work and hence, I do not recommend the publication of this manuscript in the high impact Nature Communication journal.

Reply: We are grateful for the reviewer's comments. The present research is completely different from the previous research like doping PRET system. Macrocyclic confinement single molecule PRET is rarely reported, especially accompanied by topological topography transformation. More experiments and discussions have been added in the revised manuscript. Thanks for the reviewer's comments which enabled us to eliminate the ambiguity in the original manuscript and greatly improve its quality. A point-by-point reply is attached below.

Points of concern:

1. There are several literatures regarding the excimer formation in the presence of CB8. Generally, the excimers have the emission band in the longer wavelength region w.r.t the monomer emission as well as the lifetime increases to μs . TPE-DPY molecule has

very low fluorescence yield. By forming self-assembly with CB8, the fluorescence intensity increases and the peak position at 540 nm matches well with the emission maxima of the aggregated TPE reported in the literature (J. Am. Chem. Soc. 2011, 133, 50, 20126). In the presence of CB8, TPE-DPY undergoes dimerization and shows fluorescence. The authors may have a look at the data from the above point as well.

Reply: We appreciate the reviewer's comment. According to the reviewer's advice, the experiments of the temperature-dependent delayed spectrum, time-resolved decay curves and phosphorescence spectrum under the Ar atmosphere of TPE-DPY/CB[7]/CB[8] have been performed to further prove our experimental results. The relevant results and discussions have been added to the revised manuscript. **(Page 10, lines 206-210; SI, Supplementary Fig. 36)**

2. The interpretation for phosphorescence spectrum of TPE-DPY in the presence of CB8 is mainly based on the qualitative data. The authors need to provide theoretical data to support their claim.

Reply: We appreciate the reviewer's comment. According to the reviewer's advice, the density functional theory (DFT) and time-dependent density functional theory (TDDFT) calculations have been performed to provide the theoretical data. The obtained results showed that the bromophenylpyridine units encapsulated by CB[8] can not only effectively limit its molecular motion and inhibit non-radiative transition, but also lead to enhanced intersystem crossing process, finally inducing the phosphorescence emission in an aqueous solution. The detailed calculation results and discussions are added to the revised manuscript. **(Page 12, lines 243-250, SI, Supplementary Fig. 40)**

3. The authors state that "Due to the restriction of phenyl-pyridine unit motion by cucurbituril hydrophobic cavities through host-guest complexation, the binary assembly of CB[7] or CB[8] to TPE-DPY all induced a distinct intense phosphorescent emission around 530 nm" Mechanistically how does the restriction of phenyl-pyridine unit motion facilitates triplet population? If this is correct, restriction induced by any other means (say by rigid medium) also should do the same!

Reply: We appreciate the reviewer's comment. Supramolecular assembly-based macrocyclic confinement not only effectively inhibits the non-radiative transition

caused by the disorder molecular motion and oxygen or other quenchers but also promotes the intersystem crossing process, which is different from the assembly and aggregation with no macrocyclic compounds (Chem. Asian J. 2020, 15, 3469; Angew. Chem. Int. Ed. 2023, 62, e202302751). DFT and TDDFT calculation results further confirm our experiment results and statements. The relevant experimental result and comprehensive description of macrocyclic confinement-induced phosphorescence behavior have been added to the revised manuscript. **(Page 3, lines 62-65; Page 12, lines 243-250; SI, Supplementary Fig. 40)**

5. Why no role of TPE is discussed in the formation of assembly? Only place it was discussed is for the HACD interaction at the methoxy group. But what is the electronic mechanism (other than topological change) by which NIR delayed fluorescence is induced?

Reply: We appreciate the reviewer's comment. In the assembly process, the modified TPE as an AIE molecule not only has a rigid backbone and hydrophobicity but also the aromatic conjugation matches the cavity of the cucurbituril, making it easy to direct assembly in an aqueous solution. At the same time, the electronic mechanism of NIR delayed fluorescence also has been described. More discussions have been added to the revised manuscript. **(Page 5, lines 89-93; Page 13, paragraph 1; SI, Supplementary Figs. 43, 44, 45)**

6. It was stated that "...after the injection of Ar, the lifetime of TPE -DPY/CB[7]/CB[8] aqueous solution at 540 nm was significantly increased from 59.36 μ s to 129.97 μ s (Supplementary Fig .30) due to the avoidance of the triplet electron quenching caused by oxygen, further confirming the phosphorescence properties of emission peak at 540 nm" My observation on the trace of Fig.30 says that the decay traces carry two lifetimes (one fast and other slow) and only change is in their relative amplitude contribution and the lifetime values may remain the same. Why a proper analysis is missed?

Reply: We appreciate the reviewer's comments. According to the reviewer's advice, the new experiments of time-delay curves of TPE-DPY/CB[7]/CB[8] at 540 nm under an Ar atmosphere have been performed and analyzed. The relevant experimental results and discussions have been added to the revised manuscripts. **(Page 10, lines 206-207;**

SI, Supplementary Fig. 36b)

7. The authors have qualitatively described and schematically shown the interactions and assembly formation without adequate quantitative data. How with the presence of such bulky macrocyclic groups the assembly formation is feasible?

Reply: We appreciate the reviewer's comments. According to the special property of guest molecule and cucurbituril, we deduced the assembly process of such bulky macrocyclic groups. First, the guest molecule TPE-DPY with two arms has multiple positive charges, a rigid backbone, and a flexible alkyl chain. Host molecules CB[7] and CB[8] have a rigid hydrophobic cavity, negative port, and positive outside (Chem. Rev. 2016, 116, 19, 12651–12652). Therefore, CB[8] can assembly with TPE-DPY to form a supramolecular polymer in head-to-tail binding mode, and then aggregate to nanorods through the electrostatic interactions between guest molecule and negative cucurbituril portal, and π - π stacking of guest molecule. When assembled with CB[7], it can stack to nanospheres with a stoichiometric ratio 1:4 binding mode. The assembly of TPE-DPY, CB[7], and CB[8] further increased the rigidity of supramolecular assembly and the space for stacking arrangement, finally forming multi-layered three-dimensional nanoplates. The relevant discussions have been presented in the revised manuscript. **(Page 5, lines 89-93; Page 8, lines 174-181)**

8. How do the authors justify the efficient energy transfer from one part of the complex to another part and also with HACD through space? It should have been adequately supported by the theoretical modelling justifying the placing of energy levels and their overlap function etc.

Reply: We appreciate the reviewer's comment. The relevant theoretical calculations including theoretical energy level diagrams, molecular orbitals, and the corresponding energies of PY-1/CB[8] and TPE-1/CB[7] were calculated to study the mechanism for energy transfer from one part of the complex to another part. More experimental results and discussions have been added to the revised manuscripts. **(Pages 12-13, lines 257-270, lines 277-283; SI, Supplementary Figs. 44, 45 and 48)**

9. I agree that the provided NMR data indicate the interactions, but not the mechanism of triplet to singlet energy transfer.

Reply: We appreciate the reviewer for the comment. The mechanism of triplet to singlet energy transfer in this work was discussed and confirmed by experimental and theoretical data. The relevant experiment results and discussions have been added to the revised manuscript. **(Pages 12-13, lines 250-283)**

REVIEWERS' COMMENTS

Reviewer #1 (Remarks to the Author):

In my view, the authors have answered all the questions and revised related points, and thus this revised work can be published as it is.

Reviewer #2 (Remarks to the Author):

After checked the revised manuscript, I found the authors have appropriately addressed all my concerns, as well as others. Thus I recommend the publication of this work in Nat. Common.

Reviewer #3 (Remarks to the Author):

Report on NCOMMS-23-62771A

The authors have performed additional experiments followed by theoretical calculations and addressed all the issues raised by the reviewers in the revised manuscript. I recommend the publication of this manuscript in Nature Communications.

Replies to the Reviewer

Reviewer #1 (Remarks to the Author):

General Comments: In my view, the authors have answered all the questions and revised related points, and thus this revised work can be published as it is.

Reply: We greatly appreciate the reviewer's positive comments.

Reviewer #2 (Remarks to the Author):

General Comments: After checked the revised manuscript, I found the authors have appropriately addressed all my concerns, as well as others. Thus I recommend the publication of this work in Nat. Common.

Reply: We greatly appreciate the reviewer's positive comments.

Reviewer #3 (Remarks to the Author):

General Comments: The authors have performed additional experiments followed by theoretical calculations and addressed all the issues raised by the reviewers in the revised manuscript. I recommend the publication of this manuscript in Nature Communications.

Reply: We greatly appreciate the reviewer's positive comments.